# Cardiovascular Protection with a Long-Acting GLP-1 Receptor Agonist Liraglutide: An Experimental Update

**DOI:** 10.3390/molecules28031369

**Published:** 2023-02-01

**Authors:** Collin Vandemark, Jimmy Nguyen, Zhi-Qing Zhao

**Affiliations:** Cardiovascular Research Laboratory, Mercer University School of Medicine, Savannah, GA 31404, USA

**Keywords:** angiotensin II, glucagon-like peptide 1, hypertension, heart failure, liraglutide, myocardial fibrosis

## Abstract

Angiotensin II (Ang II), a peptide hormone generated as part of the renin–angiotensin system, has been implicated in the pathophysiology of many cardiovascular diseases such as peripheral artery disease, heart failure, hypertension, coronary artery disease and other conditions. Liraglutide, known as an incretin mimetic, is one of the glucagon-like peptide-1 (GLP-1) receptor agonists, and has been proven to be effective in the treatment of cardiovascular disorders beyond adequate glycemic control. The objective of this review is to compile our recent experimental outcomes-based studies, and provide an overview the cardiovascular protection from liraglutide against Ang II- and pressure overload-mediated deleterious effects on the heart. In particular, the mechanisms of action underlying the inhibition of oxidative stress, vascular endothelial dysfunction, hypertension, cardiac fibrosis, left ventricular hypertrophy and heart failure with liraglutide are addressed. Thus, we support the notion that liraglutide continues to be a useful add-on therapy for the management of cardiovascular diseases.

## 1. Introduction

Vascular remodeling is defined as the structural changes in resistance arteries, and co-exists with the development of endothelial dysfunction and hypertension [1,2]. The mechanisms of action that have been proposed are the generation of proinflammatory mediators, reduction of endothelial nitric oxide synthase (eNOS) release, upregulation of endothelial adhesion molecules, attenuation of endothelium-dependent vasorelaxation and augmentation of extravascular cell migration [3,4]. In heart failure, morphological changes, such as cardiac hypertrophy and fibrosis, cause progressive loss of cardiomyocytes and deterioration of ventricular function through an extracellular matrix remodeling process [5,6] that includes extravascular infiltration of macrophages, differentiation of myofibroblasts, activation of TGFβ/Smads-mediated signaling and augmentation of collagen synthesis [7].

Angiotensin II (Ang II), the major bioactive peptide product of the renin–angiotensin–aldosterone system, has been extensively demonstrated to be a key player in the pathological development of cardiovascular morphology and function via altering systemic inflammatory response, blood pressure, cardiac hypertrophy, interstitial collagen deposition and tissue fibrosis, which lead to vascular dysfunction and heart failure [8]. Inhibition of Ang II-converting enzyme by angiotensin-converting enzyme inhibitors (ACEis) and blockade of Ang II AT1 receptor (AT1R) with angiotensin receptor blockers (ARBs) have emerged as an appealing therapeutic approach to improve vascular function and protect the heart against ventricular dysfunction and heart failure [9]. We have recently reported that pharmacological inhibition of AT1R inhibits inflammation, fibroblast proliferation, hypertension and myocardial fibrosis in the experimental setting of Ang II infusion [10,11]. Although ACEis and ARBs have achieved good clinical results, there might be substantial side effects such as cough, angioedema, hypotension and hyperkalemia [12,13]. Furthermore, the combination of an ACEi and ARB was associated with more adverse events without an increase in benefit [14]. Treatment with both drugs may result in a higher risk of acute kidney injury in patients with a low glomerular filtration rate [15]. The lack of an additive benefit from the combined therapy in dialysis patients with vascular disease has also been reported [16]. Therefore, the emerging adjunctive drug to be potentially used with first-line ACEis or ARBs to inhibit the progression of cardiorenal dysfunction through modulating the Ang II system still merits further investigation.

Glucagon-like peptide-1 (GLP-1) is a gastrointestinal peptide that is synthesized and secreted from intestinal enteroendocrine L cells and certain neurons within the nucleus of the solitary tract in the brainstem upon intake of nutrients such as carbohydrates, proteins and fats. However, endogenous GLP-1 is rapidly degraded, primarily by the enzyme dipeptidyl peptidase-4 (DPP-4), neutral endopeptidase and renal clearance, resulting in a half-life of approximately 2 min (Figure 1). Liraglutide is a long-acting GLP-1 receptor agonist for the treatment of type 2 diabetes. Unlike native GLP-1, liraglutide is stable and can antagonize the metabolic degradation by DPP-4 after subcutaneous administration. Recently, both preclinical studies and clinical observations have shown that the GLP-1 receptor agonist is effective at increasing GLP-1 activity and at protecting the heart against hypertension, hypertrophy and fibrosis, which is beyond the benefits achieved as an anti-diabetic agent [17,18]. We found that the cardiovascular protection from liraglutide is mediated by modulating the expression of Ang II AT1R/AT2R and GLP-1 receptor (GLP-1R) [19].

## 2. Effects of Liraglutide on Ang II AT1R, AT2R and GLP-1R

Ang II AT1R and AT2R are a class of G protein-coupled receptors expressed in the heart, blood vessels, kidney, adrenal cortex, lungs and brain. Activation of AT1R induces inflammation, endothelial dysfunction, vasoconstriction, vascular media hypertrophy, left ventricular hypertrophy and myocardial fibrosis. On the other hand, AT2R has consistently been shown to play a protective role in the attenuation of vascular smooth muscle cell proliferation, cardiovascular hypertrophy, fibrosis and cardiac dysfunction [8]. We reported that Ang II infusion or pressure overload upregulates the AT1R protein, and enhances its localization in the intracardiac vessels and myocardium, and is accompanied by a downregulation of AT2R protein and expression. These actions of Ang II were significantly blocked by liraglutide [19]. Blockade of AT1R was also associated with an inhibition of macrophage migration, fibroblast proliferation, left ventricular dysfunction and myocardial fibrosis. Consistent with data showing antagonism of AT1R, the level of AT2R protein was augmented, and its expression in the intracardiac vessels and myocardium was enhanced by liraglutide. A comparison of the beneficial effects between liraglutide and telmisartan, an AT1R blocker, on Ang II-induced changes in the expression of AT1R and AT2R [20] suggests that cardiovascular protection with liraglutide is associated with its dual actions on AT1R and AT2R (Figure 2).

GLP-1R, a member of the glucagon receptor family and G protein-coupled receptors, is expressed in the stomach, intestine, pancreatic islets, heart, lungs and kidney. It is involved in the control of blood sugar levels by enhancing insulin synthesis and release. GLP-1 synthesis begins with the same gene that produces glucagon, which is located on chromosome 6p21. The gene is transcribed into preproglucagon and then translated into protein within the rough endoplasmic reticulum. During translation, peptidase enzymes cleave a signal sequence to form proglucagon within the intestinal neuroendocrine L cells, which is in turn cleaved by prohormone convertase 1/3 into four products: GLP-1, GLP-2, glicentin and intervening peptide-2 (IP-2). GLP-1 can be further classified into GLP-1 (1–37), GLP-1 (1–36)NH2, GLP-1 (7–37) and GLP-1 (7–36)NH2 [21]. In humans, GLP-1 (7–36)NH2 accounts for 80% of GLP-1. Stimulation of GLP-1R increases adenylate cyclase through the G protein alpha subunit that increases the production of cyclic AMP (cAMP) from ATP, and subsequently activates secondary pathways, including the PKA, phosphatidylinositol-3 kinase (PI3K) and mitogen-activated protein kinase (MAPK) signaling pathways. GLP-1R also alters ion channel activity causing elevated levels of cytosolic Ca2+ that enhances insulin release from pancreatic β cells and exocytosis of insulin-containing granules [21,22]. We found that GLP-1R expression is down-regulated in response to Ang II infusion or pressure overload, whereas treatment of non-diabetic animals with liraglutide is associated with up-regulation of cardiac GLP-1R expression, indicating a direct effect of liraglutide on GLP-1R during Ang II stimulation (Figure 2). These results were further demonstrated in a murine model of heart transplantation. Administration of liraglutide from the time of transplantation upregulated GLP-1R in the heart, accompanied by an inhibition of cardiac fibrosis, inflammation and severe development of cardiac allograft vasculopathy [23].

There is a growing body of evidence that GLP-1-based therapies, independent of its blood sugar regulatory mechanism, are now widely used in the treatment of hypertension, myocardial infarction and heart failure. In our well-established rat models of Ang II osmotic infusion and pressure overload, we have reported that enhancement of GLP-1 levels using the GLP-1 agonist liraglutide or DPP-4 inhibitor linagliptin reduces cardiac fibrosis and improves myocardial function. GLP-1 preservation with its analogs has also been shown to be beneficial in a variety of animal organs such as the heart [19,20], liver [24], lungs [25] and kidneys [26] with different disease pathologies. Among the GLP-1 receptor agonists including liraglutide, semaglutide, dulaglutide, albiglutide, exenatide and lixisenatide, liraglutide is the most stable against metabolic degradation by DPP-4 with a half-life of 13 h following subcutaneous injection [17,18]. Recently, extensive research studies have shown that liraglutide exerts protection in a variety of cardiovascular diseases, such as preservation in endothelial cell function, regulation of blood pressure, attenuation of cardiomyocyte hypertrophy and myocardial fibrosis [27]. Our laboratory has a long-standing interest in the investigation of liraglutide on Ang II-induced cardiac hypertrophy, fibrosis and ventricular dysfunction, primarily focusing on the activation of oxidative stress, migration of macrophages, proliferation of myofibroblasts and deposition of collagens. To this end, this review outlines our updated experimental findings as illustrated from non-diabetic animal models of Ang II infusion and pressure overload to summarize the current understanding of the mechanisms of action underlying liraglutide-exerted cardiovascular protection.

## 3. Inhibition of Inflammation and Preservation of Vascular Function with Liraglutide

### 3.1. Effects of Liraglutide on Pro-Inflammatory Mediators and Oxidative Stress

It is well known that Ang II initiates an inflammatory cascade via enhanced overproduction of myocardial and vascular reactive oxygen species (ROS) through excessive stimulation of extracellular nicotinamide adenine dinucleotide phosphate (NADPH) oxidase. This process elicits the production of cytokines and expression of vascular adhesion molecules and induces proinflammatory and proliferative effects on endothelial cells, vascular smooth muscle cells, macrophages and cardiomyocytes. NADPH oxidase-derived ROS play a pathophysiological role in endothelial dysfunction, hypertension, inflammation, hypertrophy and fibrosis [28,29]. In a rat model of Ang II osmotic infusion, we have recently reported that Ang II stimulates NOX4/NADPH oxidase, upregulates intercellular adhesion molecule-1 (ICAM-1) and downregulates endothelial NO synthase (eNOS) on the endothelium. Furthermore, Ang II enhanced the infiltration of inflammatory cells into the tissues by stimulating the production of specific cytokines/chemokines [30]. In this regard, we found that Ang II induced the production of the potent monocyte chemoattractant protein-1 (MCP-1), and activated downstream TGFβ1 expression, macrophage migration and myofibroblast proliferation in the myocardium. Treatment with an AT1R blocker suppressed the release of inflammatory mediators such as ROS, C-reactive protein and NFκB, suggesting that AT1R is involved. We demonstrated that the dual effects of liraglutide in the antagonism of AT1R and stimulation of AT2R are associated with downregulated expression of NOX4/NADPH oxidase, ICAM-1 and MCP-1 and enhanced eNOS expression [20,30]. These anti-inflammatory and anti-oxidative effects are probably due in part to unopposed stimulation of AT2R and ACE2. Furthermore, inhibition of the extravasation of specific macrophages to the site of injury may explain the inhibitory effects of liraglutide on Ang II-induced inflammation, hypertension and perivascular fibrosis (Figure 3).

### 3.2. Effects of Liraglutide on Vascular Morphology, Perivascular Fibrosis and Hypertension

Vascular structural alterations in the development of hypertension are closely associated with the circulating levels of Ang II. The specific composition of any given artery and the changes that occur therein in hypertension mainly depend upon the intensity of Ang II stimulation [31]. We found that osmotic infusion of Ang II in rats for 4 weeks had significant effects on vascular morphology, blood pressure and hypertension development. Ang II-mediated vascular damage and hypertension with regards to endothelium-dependent vasorelaxation and vascular smooth muscle cell hypertrophy significantly occur in systemic blood vessels. Immunohistochemical studies illustrated the increased medial wall area, attenuated eNOS expression and enhanced macrophage infiltration [20,32] (Figure 3).

Perivascular fibrosis, defined as an increased amount of collagen deposition around the vessels, has been demonstrated in the cardiovascular system including arterioles and myocardial microcirculation. In rat models of Ang II infusion and thoracic aortic stenosis, we found that perivascular fibrosis was initially triggered by inflammatory cytokines, NADPH oxidase and ROS derived from the endothelium, evidenced by increased vessel wall thickness, vessel diameter and collagen deposition. Immunohistochemical staining of aortic segments also illustrated proliferation of fibroblasts and infiltration of macrophages, suggesting that extensive crosstalk among inflammatory cells, fibroblasts and the extravascular matrix actively modulates the fibrotic response in the perivascular region [10,33]. Smooth muscle cells regulate vascular resistance, and play a clear role in the development of perivascular fibrosis. Under the influence of Ang II and pressure overload, smooth muscle cells undergo phenotypic alterations from a contractile to a proliferative phenotype, leading to structural changes in the vessel walls, expressed as hypertrophy of the smooth muscle cells and fibrosis of the adventitial area [10,34]. Treatment of Ang II-induced hypertensive rats with the GLP-1R analog liraglutide or DPP-4 inhibitor linagliptin prevented medial hypertrophy, decreased wall thickness and aortic lumen size, enhanced endothelium-dependent vasorelaxation and lowered blood pressure. Such changes suggest that liraglutide plays a role in the inhibition of deleterious arterial remodeling. Compatible inhibition between liraglutide and telmisartan in the blockade of AT1R expression suggest that attenuation of Ang II-induced changes in vascular structure and function are mediated by AT1R (Figure 3).

### 3.3. Effects of Liraglutide on Endothelium-Dependent Vascular Relaxation

The endothelium plays a crucial role in the control of vascular tone and integrity by releasing a number of endothelium-derived regulating factors, such as NO, endothelium-dependent hyperpolarization factors, vasodilator prostaglandins and endothelium-derived contracting factors to maintain vascular homeostasis and preserve normal blood flow. Endothelial dysfunction has been considered a hallmark and predictor of cardiovascular disease. Damage to the vascular endothelium can affect its normal function and predispose to the development of vasoconstriction, thrombosis and hypertension through impairment of endothelium-dependent vasodilation and inflammation. Experimentally exposing blood vessels to pharmacological agents is an often selected method to assess endothelial function, i.e., NO bioavailability in different pathological conditions. In isolated mouse aortas, endothelium-dependent vasorelaxation responses to acetylcholine (ACh) is impaired by Ang II, which is associated with an augmentation of the level of NADPH oxidase expression [35]. Blockade of AT1R with telmisartan, but not the AT2 blocker PD123319, preserved aortic relaxation responses to ACh [36]. In a rat model of pressure-overload induced cardiac hypertrophy and apoptosis, endothelium-dependent vasodilation from isolated thoracic aortic rings was detected by vascular relaxation responses to the accumulative concentration of ACh. We found that thoracic aortic stenosis for 16 weeks blunted endothelium-dependent relaxation with a decreased maximal relaxation by ACh. However, the relaxing potency of the aortic rings was significantly potentiated by treatment with liraglutide. The protection was blocked by the ATP-sensitive potassium channel blocker glibenclamide and was accompanied by the downregulation of KATP channel subunit (i.e., SUR2 and SUR2) protein levels and their expression on the endothelium [10]. These data were consistent with an inhibition of AT1R by liraglutide [20]. Based on the results collected in this study, we proposed that the mechanisms of action underlying the preservation of endothelium-dependent vasodilation in response to ACh stimulation with liraglutide are associated with NO-related signaling via KATP channels.

## 4. Attenuation of Cardiac Pathological Morphology and Improvement of Cardiac Function with Liraglutide

### 4.1. Effects of Liraglutide on Cardiac Interstitial Fibrosis and Mitochondrial Morphology

Maladaptive cardiac remodeling refers to molecular, cellular and interstitial alterations in the myocardium, leading to histopathological changes in the structure, shape and function of the heart [6,7,8]. This process is associated with a poor prognosis due to ventricular dysfunction. Cardiac interstitial fibrosis as a key component of cardiac remodeling, inevitably occurs in the chronic phases of different pathological conditions including hypertension-induced ventricular pressure overload, valve defect, myocardial infarction and hypertrophic cardiomyopathy [37,38]. Stimulation of Ang II, production of reactive oxygen species, release of inflammatory mediators and deposition of collagen are implicated in the development of cardiac interstitial fibrosis. Cellular elements participating in the development of cardiac fibrosis are the vascular endothelium, smooth muscle cells, monocytic cells and resident fibroblasts [39].

Cardiac interstitial fibrosis is characterized by an increase in pathological extracellular matrix (ECM) remodeling in the myocardium, which is associated with the infiltration of macrophages, differentiation of fibroblasts and deposition of collagen. These pathological changes lead to increased matrix stiffness and abnormalities in cardiac function. Macrophages indisputably play a key role in all stages of the fibrotic process [38,39]. We found that Ang II infusion caused a significant recruitment of macrophages in the spleen. Splenectomy and blockade of AT1R reduced macrophage infiltration in the interstitial space, suggesting that macrophages are mainly accumulated in the spleen, and macrophage infiltration is mediated by stimulating AT1R [40]. Macrophages are thought to play a key role in the proliferation of the fibroblasts that produce ECM proteins responsible for the development of interstitial fibrosis. The migration of macrophages into the tissue induces fibroblasts to proliferate into myofibroblasts. We immunohistochemically investigated the relationship between infiltrating macrophages and fibroblast proliferation during Ang II infusion [41]. Macrophages produce transforming growth factor β1 (TGF-β1), which regulates the proliferation, differentiation, migration and other physiological activities of cardiac fibroblasts, and affect the repair of tissue remodeling by producing larger volumes of fibronectin and collagen along with tenascin-c and secreted protein acidic in cysteine. Furthermore, proliferated myofibroblasts can also release TGFβ1 to initiate a fibrotic process via Smads-mediated signaling pathways. This phenotype expresses alpha smooth muscle actin and develops contractile bundles. Upon activation of the TGFβ1 receptor, Smad2/3 are phosphorylated to form a heterotrimeric complex with Smad4 and subsequently bind to the TGFβ-targeted collagen genes in the nucleus. Accumulation of collagen I and III between the cardiomyocytes eventually leads to reactive interstitial fibrosis [19,40,41].

We were interested to determine the importance of each of these signaling cascades in the development of cardiac fibrosis. In our recently published studies, we tested the hypothesis that inhibition of cardiac interstitial fibrosis by the GLP-1R agonist liraglutide is mediated by fibroblast/TGFβ1/Smads signaling pathways. In both rat models of Ang II infusion and thoracic aortic stenosis, treatment with liraglutide significantly inhibited the extravascular migration of macrophages. Consistent with the time course of downregulated TGFβ1 expression, the proliferation of myofibroblasts was attenuated and phosphorylation of Smad2/3 was inhibited. Given the fact that down-regulation of Smad7 is related to enhanced expression of phosphorylated Smad2/3, inhibition of hosphor-Smad2/3 with liraglutide was primarily associated with up-regulated Smad7. Furthermore, Western blots showed that the protein levels of collagen I were reduced, and interstitial fibrosis was attenuated as confirmed by Masson’s trichrome staining [19,32,34]. These data suggest that the inhibitory effects of liraglutide on cardiac fibrosis are associated with its participation in the anti-proteolytic balance controlling ECM turnover (Figure 4).

It is known that Ang II-induced mitochondrial leakage is associated with blockage of the electron transport chain, oxidation of the membrane permeability transition pore and initiation of caspase-dependent apoptosis. These processes are mainly induced by stimulating NOX4-mediated ROS generation in mitochondria, which in turn causes opening of membrane permeability transition pore, and initiates cytochrome c/caspase-mediated apoptosis [42,43]. We recently reported that, along with increased NOX4 expression by Ang II, downregulated SIRT-3 and upregulated Bnip3 levels were detected in the myocardium [30]. Sirtuin-3 (SIRT-3) is a NAD-dependent deacetylase that is located in the mitochondrial matrix, is implicated in the regulation of mitochondrial metabolic processes and the detoxification of ROS generation. SIRT-3 depletion in hypertensive animals promotes endothelial dysfunction, vascular hypertrophy, vascular inflammation and end-organ damage [44]. Overexpression of SIRT3 in cultured cells increases respiration and decreases the production of ROS [45]. On the other hand, Bnip3, a Bcl-2/adenovirus E1B 19 kDa-interacting protein 3, modulates the permeability state of the outer mitochondrial membrane by forming homo- and hetero-oligomers inside the membrane. Upregulation of Bnip3 results in mitochondrial swelling and respiratory collapse, loss of mitochondrial membrane potential, opening of mitochondrial permeability transition pores and release of cytochrome c from isolated mitochondria in the heart, which, in turn, causes mitochondrial dysfunction and cardiac cell death [46]. Treatment with liraglutide or linagliptin reduced NOX4/ROS production and preserved mitochondrial integrity, as evidenced by reduced a number of damaged mitochondria and expression of cleaved cytochrome c. Furthermore, both interventions improved mitochondrial function, as evidenced by alterations in upregulated SIRT-3 and downregulated Bnip3 protein expression, suggesting a role of GLP-1 in the preservation of mitochondria [30].

### 4.2. Effects of Liraglutide on Mammalian Target of Rapamycin and Autophagy

mTOR (mechanistic target of rapamycin), an atypical serine/ threonine kinase, is a crucial regulator of several cellular adaptive and maladaptive processes through binding with multiple companion proteins. This atypical kinase nucleates two structurally and functionally different multiprotein complexes called mTORC1 (mTOR complex 1) and mTORC2 (mTOR complex 2). mTORC1 regulates protein synthesis, mitochondrial function, stress responses, cell proliferation and autophagy, whereas mTORC2 is involved in the regulation of cell survival and polarity [47]. The signaling network of mTORC2 is less well-characterized than that of mTORC1 in the cardiovascular system. In response to numerous extracellular and intracellular stimuli, mTOR phosphorylates a variety of substrates to regulate protein synthesis. Ribosomal protein S6 kinase beta-1, also known as p70S6K, is a main downstream substrate of mTORC1. In complexes with mTORC1, phosphorylation of p70S6K contributes to various pathologic conditions. Accordingly, pharmacological inhibition with a mTORC1 inhibitor, rapamycin, has been shown to exert protection in experimental models of inflammation, fibrosis, diabetes, obesity and cell hypertrophy [48]. Autophagy refers to a genetically programmed dynamic process that involves cell degradation, cellular component recycling and energy synthesis through the lysosomal mechanism, and helps to maintain cellular homeostasis and healthy cardiomyocytes. Disruption or deregulation of the autophagy system has been implicated in cardiovascular system dysfunction, such as myocardial infarction [49], cardiomyopathy [50] and hypertension [51]. Notably, autophagy has emerged as a key process in the pathogenesis of cardiomyopathies and heart failure. For example, excessive myocardial autophagy has been shown to contribute to Ang II-induced myocardial hypertrophy in animal models [52].

It has been demonstrated that autophagy is regulated by the mTOR/p70S6K signaling pathway. Upon activation by upstream modulators through PI3K/AKT/AMPK signaling axis, mTOR can be properly localized and activated, followed by the phosphorylation of downstream targets such as p70S6K, elF-4E binding protein (4EBP) and Unc-51-like autophagy activating kinase (ULK) [53]. In particular, phosphorylation of p70S6K has been used as a hallmark of mTOR activation, and has been correlated with autophagy inhibition within the different stages of the autophagy cascade [54,55,56,57]. Accordingly, mTOR inhibition with rapamycin [56] or knockout of mTOR [57] increases autophagy induction. Although data from animal studies and clinical observations have shown the inhibitory effects of liraglutide on cardiac fibrosis, hypertrophy and heart failure [19,33], the precise signaling mechanisms underlying its therapeutic efficacy are not fully illustrated. In our recent study, we hypothesized that the protective effect of liraglutide on a rat model of pressure-overload induced heart failure is associated with the inhibition of mTOR/p70S6K signaling and enhancement of autophagy activity. To support this hypothesis in this model, we used the mTOR inhibitor rapamycin to validate the data obtained from liraglutide treatment. mTOR/p70S6K signaling was evidenced by measuring phosphorylation of mTOR and p70S6K, while autophagy was monitored based on the ratio of LC3-II to LC3-I, expression of Beclin-1 and protein levels of p62 [11]. During autophagy induction, soluble LC3-I in the cytoplasm can be transformed into LC3-II on autophagosome membranes [58]. Therefore, the conversion of LC3-I or production of LC3-II has been commonly used to reflect autophagic activity [59]. Beclin 1 is a key molecule in the control of autophagic activity by regulating post-translational modification and protein–protein interactions to promote autophagosome maturation. Downregulation of Beclin 1 contributes to the pathogenesis of cardiac hypertrophy and heart failure by defecting autophagic formation [60]. p62 is a marker of the degradation phase of autophagosomes, and is negatively correlated with autophagic activity. As expected, liraglutide significantly enhanced autophagy activity, as shown by an increase in the LC3-II/LC3-I ratio and in Beclin-1 expression as well as a decrease in p62 expression. Compatible modulations of the LC3-II/LC3-I ratio, Beclin-1 and p62 with rapamycin suggested that the restoration of autophagy by liraglutide is potentially mediated through a GLP-1/mTOR/p70S6K axis-dependent autolysosome signaling pathway [60]. Our data were consistent with a recent study showing that liraglutide exerts renoprotection in a rat remnant kidney model of chronic renal failure by promoting autophagic flux [61].

### 4.3. Effects of Liraglutide on Cardiac Hypertrophy

Pathological cardiac hypertrophy represents one of the important cardiovascular disorders. Initially, it is characterized by the increased mass and size of cardiomyocytes along with the formation of new sarcomeres to normalize ventricle wall stress and allow normal cardiovascular function, known as an essential compensatory mechanism [62]. However, the hypertrophied heart eventually decompensates, ultimately leading to left ventricle dilation and heart failure. There are two types of cardiac hypertrophy based on the time course of pathogenesis. Concentric hypertrophy is caused by pressure overload with mechanical stiffening and an increased filling resistance, contributing to diastolic dysfunction with preserved ejection fraction, while eccentric hypertrophy is characterized by enlarged ventricular chamber size and marked cardiomegaly due to volume overload, attributed to the decline in systolic function with a reduced ejection fraction. Under the conditions of pressure overload, concentric hypertrophy may develop eccentric hypertrophy due to progressive loss of contractile function. Ultimately, for a favorable prognosis, the management of left ventricular hypertrophy is always the object of therapeutic intervention [63].

Ang II plays an important role in the development of myocardial hypertrophy, which is mainly caused by sustained increased afterload during the development of hypertension. As we reported recently, Ang II infusion or thoracic aortic stenosis increased the size of cardiomyocytes, and induced left ventricular hypertrophy in vivo [10,34]. The mechanism of action underlying the pathophysiology of cardiac hypertrophy is a complex process. Activation of the AT1R enhances protein synthesis and fetal gene expression, including atrial natriuretic peptide (ANP), brain natriuretic peptide (BNP), β-myosin heavy chain (βMHC) and reactive oxygen species (ROS) production. Dysregulation of various signaling pathways including the mitogen-activated protein kinase (MAPK), protein kinase C (PKC), nuclear factor of activated T cells (NFAT), phosphoinositide 3-kinase (PI3K)/protein kinase B (PKB/Akt), mammalian target of rapamycin (mTOR) and insulin-like growth factor (IGF)-1 pathways, may contribute to the progression of cardiac hypertrophy [52,53,64]. In a rat model of abdominal aortic constriction (AAC), we found that at the cellular level, AAC caused a significant increase in oxidative stress, migration of macrophages and proliferation of myofibroblasts. At the protein level, the expression of TGFβ1/Smads was upregulated and the deposition of collagen was enhanced. At 16 weeks of AAC, hematoxylin and eosin staining revealed a significant increase in the size of cardiomyocyte in cross-section, consistent with an increased heart/body weight ratio. Masson’s trichrome staining confirmed abundant perivascular and interstitial collagen deposition. Treatment with liraglutide and telmisartan significantly reduced these AAC-induced morphological changes. Blockade of AT1R with telmisartan showed a comparative level of protection as seen in the liraglutide group, suggesting that these beneficial effects are associated with inhibition of AT1 receptor-mediated events in cardiac hypertrophy. Furthermore, liraglutide up-regulated plasma GLP-1 expression in the 16 weeks of AAC in a rat model of Ang II osmotic infusion [10,11]. The preservation of morphology with liraglutide was consistent with its beneficial effects on cardiac function.

### 4.4. Effects of Liraglutide on Cardiac Function and Heart Failure

In response to left ventricular stress or pressure overload, the dynamic process of cardiac remodeling causes many structural and functional alterations, such as ventricular hypertrophy, ventricular fibrosis, ventricular contractile dysfunction/dilation, cardiomegaly and other changes. Initial ventricular remodeling as an adaptive response aims to maintain stable cardiac function, and typically manifests as concentric remodeling. Over time, a progressive increase in ventricular volumes and decline in ventricular function result in eccentric remodeling, leading to progressive ventricular dysfunction and heart failure [65]. We and others have demonstrated that the cellular elements involved include endothelial cells, macrophages and fibroblasts. This happens as a result of sustained Ang II activation, propagation of myocardial injury and hemodynamic inefficiency, which activate multiple molecular pathways including the AKT/mTOR, MEK/ERK, JAK/STAT and TGF-β1/Smad pathways [11,64]. In a rat model of AAC, we demonstrated a significant reduction in ejection fraction following 16 weeks of chronic hemodynamic overload. The data clearly showed that the significant ventricular global dilatation and systolic dysfunction with pressure overload elicits Ang II stimulation, and promotes the development of eccentric hypertrophy [11]. In this regard, we and others have shown that both ACEis and ARBs have significant effects on the reversal of adverse cardiac remodeling. To demonstrate the efficacy of liraglutide in this heart failure model, we treated the animals with a subcutaneous injection of liraglutide on a daily basis for 16 weeks, and found that liraglutide significantly reduced chamber dilatation and attenuated myocardial remodeling. Left ventricular function was significantly improved as measured by echocardiography, including left ventricular internal diameter at diastole, left ventricular internal diameter at systole, left ventricular posterior wall diameter, fractional shortening and ejection fraction. Consistent with these global measurements, invasive hemodynamic measurements also showed reduced mean arterial pressure and left ventricular end-diastolic pressure, and increased left ventricular systolic pressure and maximum positive (dp/dtmax) values of the first derivative of left ventricular pressure [10]. Although we cannot conclude whether liraglutide has a direct inotropic effect on the heart, the comparative inhibition in lipid peroxidation, hypertrophy and fibrosis as demonstrated between liraglutide and telmisartan in these studies suggests that these beneficial effects are associated with a blockade of AT1R [15]. Furthermore, 16 weeks of pressure overload reduced plasma GLP-1 levels and downregulated GLP-1R expression in the myocardium, suggesting a decompensated status of GLP-1 in the response to a long period of pressure overload. Liraglutide treatment increased the plasma concentration of GLP-1 and the expression of GLP-1R in the myocardium [11]. These data were consistent with other reports showing that enhanced GLP-1R protein and plasma GLP-1 concentrations are associated with improved ejection fraction [66]. Therefore, GLP-1 alteration with liraglutide may serve as an alternative compensatory mechanism to protect the heart against ventricular dysfunction. We have also reported that enhancement of GLP-1 levels with a GLP-1 analog or DPP-4 inhibitor after angiotensin II infusion augments the expression of GLP-1 receptor and inhibits cardiac fibrosis [11,19].

## 5. Future Prospect of Liraglutide Investigation

The current experimental evidence supports beneficial effects of liraglutide on adverse cardiovascular events [67]. However, the following potential future studies might need to be considered: (1) most of our studies were performed using non-diabetic models that showed an improvement in cardiovascular function. So far, it has not fully addressed whether liraglutide exerts protection on Ang II- or pressure overload-induced diastolic dysfunction with preserved ejection fraction or systolic dysfunction with reduced ejection fraction in diabetic models. (2) The conventional treatment for patients with cardiac diastolic or systolic dysfunction are ACEis, ARBs or β-blockers followed by diuretics. However, there are also some clinical observations showing a lack of protection in the prevention of heart failure with these interventions [68]. Currently, no unique drugs are recommended as a first-line therapy to protect the heart against Ang II- or pressure overload-induced heart failure. It remains unclear from the clinical trials whether the lack of full recovery from heart failure with these interventions could be further improved if liraglutide and ACEi or ARBs are combined [69]. (3) In our studies, enhancement of GLP-1 levels with exogenous supplementation of liraglutide or endogenous inhibition of GLP-1 degradation with DPP-4 inhibitor linagliptin inhibited Ang II-induced cardiac fibrosis. However, it is warranted to check whether combining the two drugs would confer additional benefits or negate each other’s beneficial effect. (4) Furthermore, the time and route of liraglutide administration varied across studies, and it is not known whether there are differences between the efficacy of short-term liraglutide infusion and long-term oral administration of liraglutide at different doses in the inhibition of cardiac hypertrophy and fibrosis. Finally, (5) we verified that cardiac protection with liraglutide is mediated by upregulating the expression of GLP-1R. However, direct evidence showing an association between liraglutide treatment and GLP-1 receptor expression needs to be further explored using a model of GLP-1 deficiency under the conditions of Ang II infusion or pressure overload. Therefore, the illustration of cardioprotection with liraglutide as discussed above may further provide experiment-based mechanisms of action, and help to explain the conflicting data between the experimental outcomes and clinical investigations from different clinical trials.

## 6. Conclusions

The existing literature appears to hold promise that GLP-1R agonists and DPP-4 inhibitors might exert cardioprotective effects beyond glycemic control in patients with hypertension and heart failure. Liraglutide, as the first long-acting GLP-1R agonist, is the best-selling drug for the treatment of type 2 diabetes mellitus or obesity. The numerous beneficial effects of liraglutide render this drug as an interesting candidate for the development of pharmacotherapies to treat obesity, diabetes and neurodegenerative disorders. Although the precise mechanisms of action underlying cardioprotection with liraglutide are still being explored, this review summarized the updated experiments, primarily conducted in our laboratory, by focusing on its effects on the regulation of vascular function, hypertension, cardiac hypertrophy and fibrosis in response to Ang II stimulation and pressure overload using non-diabetic animal models. We highlighted that liraglutide at the dose selected for glycemic control effectively inhibits oxidative stress, vascular endothelial dysfunction, hypertension, cardiac fibrosis, left ventricular hypertrophy and heart failure. These results may open a window of opportunity to explore the interventional strategies for other GLP-1 receptor agonists (e.g., semaglutide, dulaglutide and albiglutide) or sodium–glucose cotransporter 2 inhibitors (e.g., dapagliflozin and empagliflozin) in cardioprotection independent of improved glycemic control, where hypertension, cardiac hypertrophy and fibrosis are the main therapeutic targets.

## Figures and Tables

**Figure 1 molecules-28-01369-f001:**
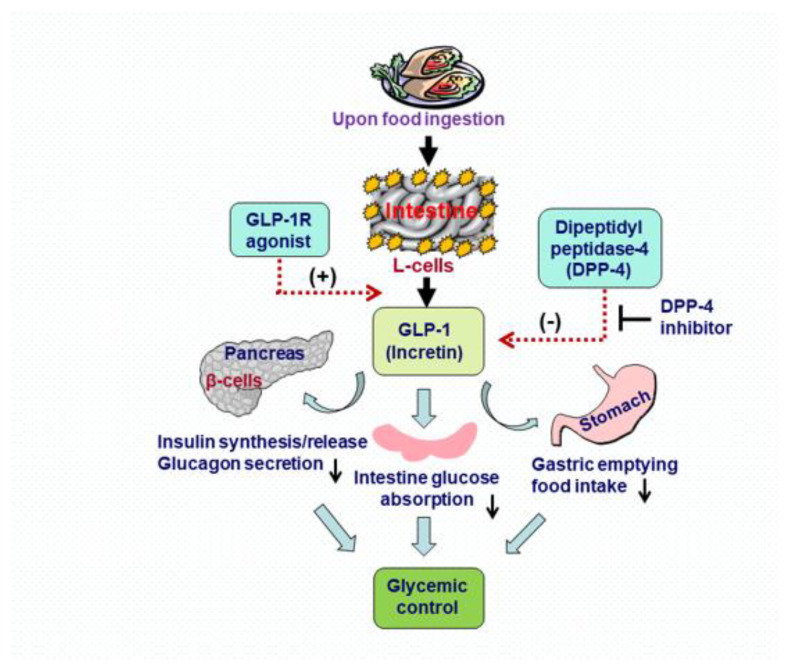
Glycemic control of glucagon-like peptide-1 (GLP-1). GLP-1 is an incretin glucoregulatory hormone released from the L cells of the intestine in response to food intake. When plasma glucose levels are elevated, GLP-1 stimulates the β cells of the pancreas to release insulin, decreases glucagon secretion from the intestine via acting on the GLP-1 receptor, and slows gastric emptying and food intake (downward arrows). Endogenous GLP-1 is extremely susceptible to the catalytic activity of the proteolytic enzyme dipeptidyl peptidase-4 (DPP-4), and can be rapidly metabolized within a few minutes once secreted. Enhancement of GLP-1 levels by an exogenous supply of GLP-1 receptor agonist (+) or inhibition of endogenous GLP-1 degradation with DPP-4 inhibitor (−) shows a significant beneficial effect in the control of blood glucose and the development of obesity in type 2 diabetes.

**Figure 2 molecules-28-01369-f002:**
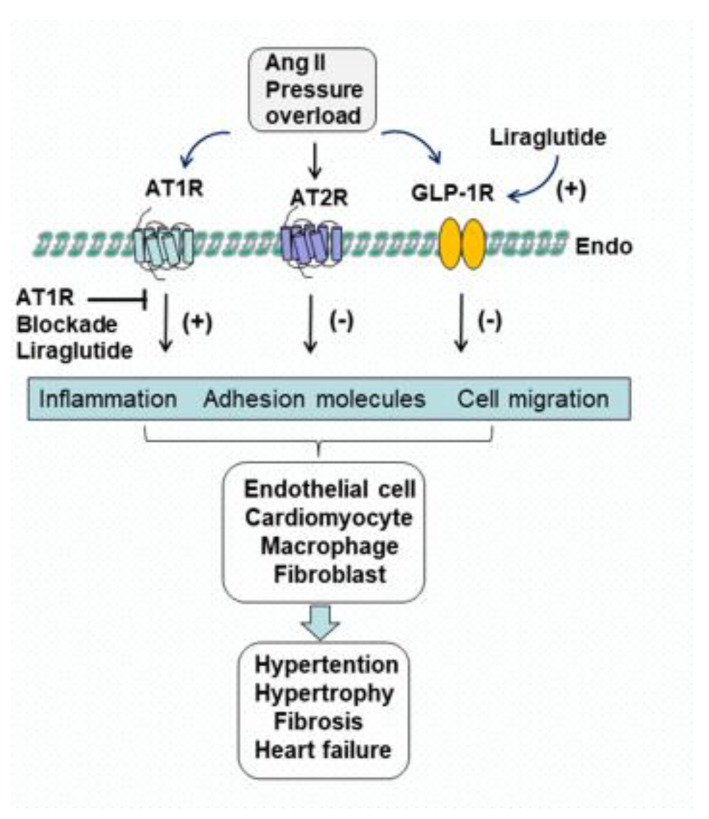
The signaling pathways involved in cardiovascular regulation under activation of the two Ang II receptor subtypes, AT1 and AT2 receptor (AT1R and AT2R) as well as GLP-1 receptor (GLP-1R). Ang II infusion or pressure overload elicits inflammatory responses, upregulates adhesion molecular expression on the vascular endothelium (Endo) and promotes extravascular cell migration by stimulating AT1R (+), further resulting in hypertension, cardiac hypertrophy, fibrosis and heart failure. There is functional cross-talk between AT1R and AT2R in these signaling pathways. Moreover, the activation of the AT2R causes vasodilatory, anti-inflammatory, antiproliferative and antifibrotic effects by eliminating the AT1R-mediated responses (−), evidenced by reduced hypertension and improved cardiac function. The GLP-1R agonist liraglutide activates GLP-1R (+), and protects the cardiovascular system, yielding beneficial effects as compared to blockade of AT1R (-), suggesting that the protection is potentially mediated by inhibiting AT1R via GLP-1R. Endo: endothelium.

**Figure 3 molecules-28-01369-f003:**
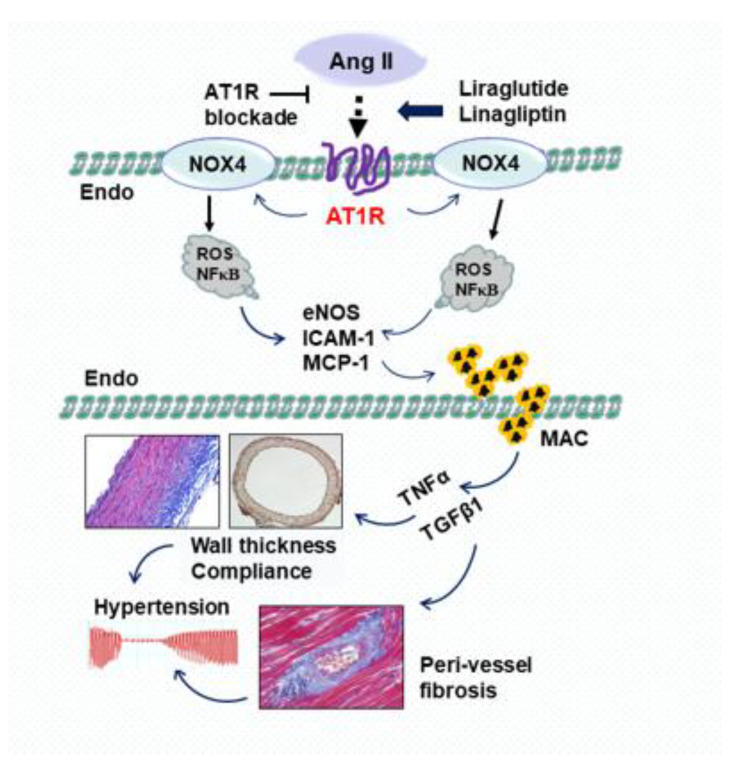
The signaling pathways proposed in vascular protection by preserving GLP-1 using the GLP-1R agonist liraglutide and DPP-4 inhibitor linagliptin as well as the AT1R blockade. Ang II activates NOX4/NADPH oxidase expressed on the endothelium by stimulating AT1 receptor. The increased production of reactive oxygen species (ROS) and NFκB by NOX4 causes eNOS uncoupling, enhances the expression of intercellular adhesion molecule-1 (ICAM-1) and monocyte chemoattractant protein-1 (MCP-1), and facilitates the migration/infiltration of monocytes/macrophages (MAC). TNFα and TGFβ1 released from macrophages increase vascular wall thickness, decrease vascular compliance and induce perivascular fibrosis, resulting in endothelial dysfunction and hypertension. Upregulating GLP-1 levels with liraglutide/linagliptin or blocking AT1R inhibits the cascades of the Ang II/NOX4/ROS signaling pathways, and protects vascular function via inhibiting Ang II-induced vascular remodeling. Endo: endothelium.

**Figure 4 molecules-28-01369-f004:**
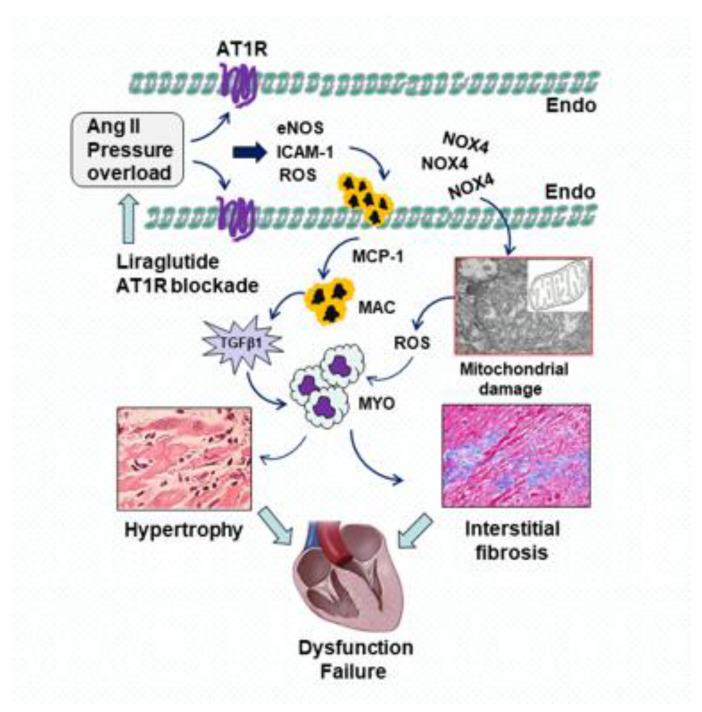
Potential signaling pathways involved in cardioprotection by GLP-1 receptor (GLP-1R) agonist and AT1 receptor (AT1R) blockers on Ang II and pressure overload-induced cardiomyocyte hypertrophy, tissue fibrosis and cardiac dysfunction. Ang II or pressure overload induces eNOS uncoupling and enhances the expression of intercellular adhesion molecule-1 (ICAM-1) through activation of AT1R. Furthermore, the production of reactive oxygen species (ROS) from activated endothelium and mitochondria facilitates the migration of macrophages (MACs) via overexpressed monocyte chemoattractant protein-1 (MCP-1). TGFβ1 released from interstitial macrophages promotes the proliferation of myofibroblasts (MYO), leading to myocardial morphological change and heart failure. Blockade of AT1R with liraglutide or telmisartan reduces the expression of AT1R, inhibits the cascade of the AT1R/NOX4/ROS signaling pathways, and thereby protects the heart against Ang II-induced myocardial remodeling.

## Data Availability

Not available.

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
