# Peer review of "Cardiovascular Protection with a Long-Acting GLP-1 Receptor Agonist Liraglutide: An Experimental Update"

_molecules, 2023, doi:10.3390/molecules28031369_

Round 1

Reviewer 1 Report

The MS is well written, structured and the figures are understandable. Watch for typos, spaces, and consistency in spelling abbreviations, e.g., ACh or Ach? Section 2 Cite in the text some studies on liraglutide and the dual effect on the AT1R and AT2R. Briefly mention some other in vivo and in vitro studies on semaglutide, dulaglutide, and albiglutide in cardiovascular disease and compare them with studies on liraglutide.

Author Response

Responses to Reviewer #1

The MS is well written, structured and the figures are understandable. Watch for typos, spaces, 

and consistency in spelling abbreviations, e.g., ACh or Ach? Section 2 Cite in the text some studies on liraglutide and the dual effect on the AT1R and AT2R. Briefly mention some other in vivo and in vitro studies on semaglutide, dulaglutide, and albiglutide in cardiovascular disease and

compare them with studies on liraglutide.

Responses:  Spelling abbreviations have been carefully checked. Potential application of other GLP-1 receptor agonists as mentioned above in cardioprotection have been addressed (please see lines in text 525-529), thank you!

Reviewer 2 Report

The review is well written and includes primarily a summary of their own work, however it is clearly stated in the review when they are referring to their own work and results which helps to minimise any bias in the presentation or interpretation of the literature summary. Minor comments below, 

  • The authors may consider adding one small paragraph comparing other co-treatment with other insulin sensitising drugs such as Dapagliflozin or empagliflozin. How do these compare to liraglutide considering that the general theme of the review is that liraglutide has positive effects that are independent of improved glycaemic control.
  • Line 23. Correct to co-exists instead of co-existed
  • Line 135. Correct to enhancement instead of enchantment
  • Line 233. Correct font change
  • Line 388. Correct font change.
  • Line 456. Correct font change.
